The cnidarian parasite Ceratonova shasta utilizes inherited and recruited venom-like compounds during infection

Americus Benjamin 1
Hams Nicole 5
http://orcid.org/0000-0001-8939-0057 Klompen Anna M. L. 2
http://orcid.org/0000-0002-6452-6154 Alama-Bermejo Gema 1 3
http://orcid.org/0000-0002-1679-4904 Lotan Tamar 4
http://orcid.org/0000-0001-8183-9390 Bartholomew Jerri L. 1
http://orcid.org/0000-0002-7180-6672 Atkinson Stephen D. 1 stephen.atkinson@oregonstate.edu
1 Department of Microbiology, Oregon State University , Corvallis, Oregon , United States of America
2 Department of Ecology and Evolutionary Biology, The University of Kansas , Lawrence, Kansas , United States of America
3 Institute of Parasitology, Biology Centre, Czech Academy of Sciences , České Budějovice , Czech Republic
4 Marine Biology Department, The Leon H. Charney School of Marine Sciences, University of Haifa , Haifa , Israel
5 Columbia River Fish and Wildlife Conservation Office, U.S. Fish and Wildlife Service , Vancouver, Washington , United States of America
Uversky Vladimir
Electronic publication date: 2021 Dec 15
Publication date: 2021
Volume: 9
Electronic Location ID: e12606
Received 2021 Jul 2; Accepted 2021 Nov 16
Copyright: © 2021 Americus et al.
Copyright year: 2021
Copyright holder: Americus et al.
License: This is an open access article distributed under the terms of the Creative Commons Attribution License, which permits unrestricted use, distribution, reproduction and adaptation in any medium and for any purpose provided that it is properly attributed. For attribution, the original author(s), title, publication source (PeerJ) and either DOI or URL of the article must be cited.
License URL: https://creativecommons.org/licenses/by/4.0/

Keywords: Toxin, Venom, Myxozoa, Cnidaria, Nematocyst, Proteome, Transcriptome, Myxospore

Funding: United States-Israel Binational Agricultural Research and Development (BARD) IS-5001-17C United States-Israel Binational Science Foundation (BSF), Jerusalem, Israel 2019063 This research was supported by research grant no. IS-5001-17C from the United States-Israel Binational Agricultural Research and Development (BARD) Fund and research grant no. 2019063 from the United States-Israel Binational Science Foundation (BSF), Jerusalem, Israel. The funders had no role in study design, data collection and analysis, decision to publish, or preparation of the manuscript.

==============================
Background

Cnidarians are the most ancient venomous organisms. They store a cocktail of venom proteins inside unique stinging organelles called nematocysts. When a cnidarian encounters chemical and physical cues from a potential threat or prey animal, the nematocyst is triggered and fires a harpoon-like tubule to penetrate and inject venom into the prey. Nematocysts are present in all Cnidaria, including the morphologically simple Myxozoa, which are a speciose group of microscopic, spore-forming, obligate parasites of fish and invertebrates. Rather than predation or defense, myxozoans use nematocysts for adhesion to hosts, but the involvement of venom in this process is poorly understood. Recent work shows some myxozoans have a reduced repertoire of venom-like compounds (VLCs) relative to free-living cnidarians, however the function of these proteins is not known.

Methods

We searched for VLCs in the nematocyst proteome and a time-series infection transcriptome of Ceratonova shasta, a myxozoan parasite of salmonid fish. We used four parallel approaches to detect VLCs: BLAST and HMMER searches to preexisting cnidarian venom datasets, the machine learning tool ToxClassifier, and structural modeling of nematocyst proteomes. Sequences that scored positive by at least three methods were considered VLCs. We then mapped their time-series expressions in the fish host and analyzed their phylogenetic relatedness to sequences from other venomous animals.

Results

We identified eight VLCs, all of which have closely related sequences in other myxozoan datasets, suggesting a conserved venom profile across Myxozoa, and an overall reduction in venom diversity relative to free-living cnidarians. Expression of the VLCs over the 3-week fish infection varied considerably: three sequences were most expressed at one day post-exposure in the fish’s gills; whereas expression of the other five VLCs peaked at 21 days post-exposure in the intestines, coinciding with the formation of mature parasite spores with nematocysts. Expression of VLC genes early in infection, prior to the development of nematocysts, suggests venoms in C. shasta have been repurposed to facilitate parasite invasion and proliferation within the host. Molecular phylogenetics suggested some VLCs were inherited from a cnidarian ancestor, whereas others were more closely related to sequences from venomous non-Cnidarian organisms and thus may have gained qualities of venom components via convergent evolution. The presence of VLCs and their differential expression during parasite infection enrich the concept of what functions a “venom” can have and represent targets for designing therapeutics against myxozoan infections.

Introduction

Cnidarians are the earliest diverging extant venomous animals (Jouiaei et al., 2015). This ancient and taxonomically diverse phylum of aquatic invertebrates is characterized by a unique organelle, the nematocyst. Inside each nematocyst is a coiled and eversible tubule, which can be discharged to deliver a venomous sting (Beckmann & Özbek, 2012). Venom is a cocktail of toxic compounds, stored inside the unfired inverted tubule and nematocyst (Lotan et al., 1995). When a cnidarian encounters chemical and physical cues, the nematocyst is triggered and the tubule everts, injecting the venom into the host or the immediate environment. In Cnidaria, most venom constituents are proteins, broadly comprising enzymes, pore-forming toxins, and neurotoxins (Jouiaei et al., 2015; Rachamim et al., 2015; Podobnik & Anderluh, 2017; D’Ambra & Lauritano, 2020). These compounds are useful in immobilizing and digesting prey and deterring predators (Orts et al., 2013). Note that for the many taxa where functional studies have not been undertaken, candidate venoms can be inferred from sequence or structural homology; we refer to these here as venom-like compounds (VLCs).

Nematocyst structure and venom composition vary among the three main clades of Cnidaria: Anthozoa and Medusozoa, which are predominantly free-living, and Endocnidozoa, which contains the enigmatic and parasitic Myxozoa (Rachamim et al., 2015; Kayal et al., 2018; Americus et al., 2020). Myxozoans require two hosts and alternate between vertebrate-infective actinospores and invertebrate-infective myxospores (Okamura, Gruhl & Bartholomew, 2015). Anthozoa (sea anemones, corals, etc.) and Medusozoa (medusa-bearing species) have a wide variety of nematocyst morphotypes both within and between species (Fautin, 2009). The few myxozoan nematocysts described have tubules with uniform thickness along their lengths, lack spines (Uspenskaya, 1982; Östman, 2000; Ben-David et al., 2016) and have sealed tips (Ben-David et al., 2016; Piriatinskiy et al., 2017; Americus et al., 2020). These are morphologically similar to the atrichous isorhiza-type nematocysts of medusozoans (Cannon & Wagner, 2003; Americus et al., 2020). Rather than being employed for predation or defense, myxozoans use nematocysts for adhesion to hosts (Ben-David et al., 2016; Foox et al., 2015).

When a myxozoan actinospore encounters physical and chemical cues produced by a potential host, the tubule everts, penetrating the host epithelium and anchoring the parasite (Kallert et al., 2007). In some “freshwater” clade Myxobolus spp. the tubule then contracts, pulling the apical end of the spore in contact with the host. Some Myxobolus spp. also discharge nematocyst contents through lateral pores in the tubule, though this is notably absent in the model “marine” clade myxozoan, Ceratonova shasta (Ben-David et al., 2016). Once attached, a multinucleate/multicellular sporoplasm migrates out of the spore and into the host, beginning the infection (Bjork & Bartholomew, 2010; Kallert et al., 2007).

Myxozoans may have lost the venom proteins of their free-living relatives in a genome reduction with the shift to parasitism (Piriatinskiy et al., 2017). This is evidenced by simplifications of morphology and function—primarily for attachment to a host (Kallert et al., 2007). But these reductions differ among myxozoan taxa, for example a transcriptome analysis of Myxobolus pendula cysts in fish identified 49 VLCs (Foox et al., 2015), whereas searches for VLCs in the C. shasta nematocyst proteome (Piriatinskiy et al., 2017) identified only a single venom-like domain in a protein of unknown function, which also supports the hypothesis that myxozoan nematocysts do not contain venoms. There is, however, some evidence for myxozoans having VLCs localized outside nematocysts (Hartigan et al., 2021). This is known from free-living cnidarians, where venoms may be secreted from glands in the ectoderm and pharynx for prey immobilization and digestion (Moran et al., 2012, 2013; Zhang et al., 2003).

Hartigan et al. (2021) recently identified VLCs in transcriptomes and proteomes from the myxozoans Buddenbrockia plumatellae, Myxobilatus gasterostei and Sphaerospora elegans, the semi-parasitic endocnidozoan Polypodium hydriforme, and the free-living medusozoan Calvadosia cruxmelitensis. The myxozoans had only 1/3 the venom diversity relative to the free-living species, whereas P. hydriforme had 2/3 the venom diversity of the free-living species, which is consistent with its intermediate phylogenetic position between free-living cnidarians and Myxozoa (Kayal et al., 2018). Some VLCs in the endocnidozoan datasets appear inherited from free-living cnidarians, with phylogenies correlating with established taxonomy. Other VLCs were more closely related to those from venomous animals outside of Cnidaria, suggesting convergent evolution and a recruitment of venoms for other physiological roles.

In this study, we continue the search for VLCs in myxozoans. We reexamine the proteome of C. shasta nematocysts (Piriatinskiy et al., 2017) and a time-series of infection transcriptome (Barrett & Bartholomew, 2021) to identify candidate VLCs and infer their functions. We hypothesize that C. shasta has retained a venom arsenal from a cnidarian ancestor and predict that adaptation to a parasitic lifestyle has led to reduction in venom diversity relative to extant free-living species. We further hypothesize that the C. shasta VLCs are highly dissimilar from those of extant free-living Cnidaria but will have retained recognizable functional “venom” domains, as is the case in myxozoan ribosomal (Evans et al., 2010) and mitochondrial genes (Takeuchi et al., 2015).

Materials and Methods

We used four parallel approaches to identify candidate sequences of venom-like compounds (VLCs): (1) structural phylogenetics, (2) BLAST, (3) HMMER, and (4) ToxClassifier (Fig. 1). We considered sequences that were positive by at least three methods to be putative VLCs. In the case of multiple VLCs from a single Trinity read cluster, we selected the longest sequence for downstream analysis. Sequences in the infection transcriptome and their identical translations in the nematocyst proteome were considered to be single VLCs.

Figure 1 Bioinformatic pipeline for identification and characterization of genes coding for venom-like compounds.

Functional prediction by structural modeling

To predict the tertiary structure of the C. shasta nematocyst peptides (Piriatinskiy et al., 2017), we submitted them to the Phyre2 server (Kelley et al., 2015) with the Normal modelling mode. Protein function was assigned based on homology to known structures in the Protein Data Bank (PDB) or families in the Structural Classification of Proteins database (SCOP; accessed March 2019). Results were parsed by the presence of structurally similar sequences whose functions contained keywords (protease, inhibitor, toxin, venom). We submit structures to the Dali Protein Structure Comparison Server all-against-all structure comparison tool (Holm & Laakso, 2016) for tree building (Supplemental File 2). We did not perform this analysis on the infection transcriptome (42,542 transcripts) due to the uncertainties introduced in peptide prediction, and limitations of manual analysis.

Homology search

We manually curated a database of venom and toxin proteins and transcripts from publicly available venom transcriptomes and proteomes: box jellyfish Alatina alata (Lewis Ames et al., 2016), pore-forming proteins from several free-living cnidaria (Podobnik & Anderluh, 2017), putative toxin transcripts from the acrorhagi of Anthopleura elgantissima sea anemones (Macrander, Brugler & Daly, 2015), venom transcripts from the tentacles, mesentery, and column of Anemonia sulcata, Heteractis crispa, and Megalactis griffithsi anemones (Macrander, Broe & Daly, 2016), venom protein sequences from the box jellyfish Chironex fleckeri (Brinkman et al., 2012), protein contents of the nematocysts of the sea anemone Nematostella vectensis (Moran et al., 2013), and the UniProt manually-annotated animal venom and toxin protein database (downloaded January, 2019). Our custom database consisted of 31,798 transcripts and 7,129 proteins. We screened the C. shasta nematocyst proteome and infection transcriptome against this database using BLAST searches (e-value cutoff 1e−5). In parallel, we screened the same C. shasta sequences against the cnidarian venom-like HMM database from Klompen et al. (2020), with a less stringent e-value cutoff (1e−3) to allow comparison with similar work by Klompen et al. (2020) on Cerianthid transcriptomes.

ToxClassifier

We input the nematocyst proteome and translated infection transcriptome into the web-based version of the machine learning tool ToxClassifier (Gacesa, Barlow & Long, 2016). We considered sequences deemed as either “toxin” or “dubious” by one or more classifiers as positive hits for selecting VLCs.

Phylogenetics of putative venom-like compounds

We constructed phylogenetic trees of the putative VLCs identified by at least three screening methods. We used BLASTP to identify and download homologous sequences from GenBank (queried November 2020). To test our hypothesis of venom inheritance from a cnidarian ancestor, we included sequences from Polypodium hydriforme (the non-myxozoan member of Endocnidozoa), which is an evolutionary intermediate between Myxozoa and free-living Cnidaria, and which has nematocysts that are structurally similar to the venomous holotrichous isorhiza of free-living cnidarians (Ibragimov & Raikova, 2004). We also included sequences from Medusozoa (sister to Endocnidozoa), more distantly related anthozoans, and rooted the tree using homolgous sequences from Porifera. Sequences were aligned with MUSCLE (Edgar, 2004), then trimmed, and the best alignment models selected using MEGA-X (version 10.2.5, Kumar et al., 2018). We constructed maximum likelihood trees with IQ-TREE (Version 2.1.2, Minh et al., 2020) using 1,000 bootstrap replicates. We used FigTree (version 1.4.4, http://tree.bio.ed.ac.uk/) to visualize all trees and we included ultrafast bootstrap support values >50. We compared the topology of our single-gene trees to phylogenomic trees constructed from whole-transcriptome comparison (Kayal et al., 2018; Chang et al., 2015), to ascertain whether the gene was inherited from a common medusozoan ancestor or acquired some other way. The untrimmed transcripts for each tree were input into the NCBI conserved domains search tool (Marchler-Bauer et al., 2015) for domain visualization.

Analysis of time-series expression during infection in fish

The time-series infection data were obtained from Barrett & Bartholomew (2021), (NCBI BioProject PRJNA694439), from susceptible rainbow trout that were exposed to C. shasta genotype IIR for 24 h, then held for 21 d. Gills were sampled at 1 d post-exposure (dpe), and intestines were sampled at 7, 14, and 21 dpe. Fish exposures, RNA-extraction and sequencing are described in detail in Barrett & Bartholomew (2021). RNA-Seq data were generated using an Illumina HiSeq 3000 with 100-bp single-end runs. We used BBDuk (version 38.11, https://sourceforge.net/projects/bbmap/) to remove reads with >20 bases with PHRED score <20, reads containing homopolymers >50 bases, and reads with ≥20 bases overlapping with Illumina Adapters. The remaining “cleaned” reads from all timepoints were pooled, host-filtered, and assembled into a reference transcriptome using a published pipeline (Alama-Bermejo et al., 2020). A newer version of the host genome was used for mapping during host contamination filtration (NCBI Assembly GCF_002163495.1). We retained reads that mapped to the C. shasta genome, and neither C. shasta nor host genomes. We used Transdecoder (version 5.5.0, Haas et al., 2013), with default settings and the—single_best_orf option to generate amino acid sequences for downstream analyses. The translated transcriptome and the nematocyst proteome (Piriatinskiy et al., 2017) were annotated using HMMER (version 3.3, hmmer.org) to compare against the Pfam database (accessed December 2020).

To generate a read count matrix for transcripts at each timepoint, we used Salmon (version 0.10.0, Patro et al., 2017) to pseudo-align reads from each timepoint to the time-series transcriptome. We input read counts into DESeq2 (version 3.12, Love, Huber & Anders, 2014) using tximport (Soneson, Love & Robinson, 2015). To correct for parasite replication within the host, we used the estimateSizeFactors function in DESeq2 to normalize transcript counts based on two myxozoan single-copy genes: glyceraldehyde-3-phosphate dehydrogenase and eukaryotic translation elongation factor 2 (Kosakyan et al., 2019). We analyzed expression in terms of read count per gene rather per isoform to better distinguish functional roles within the host. Read counts underwent variance-stabilizing transformation prior to downstream analysis. To distinguish co-expressed genes, we input normalized read counts into the Short Time-series Expression Miner (STEM; version 1.3.13, Ernst & Bar-Joseph, 2006) with default settings and without normalization and “0” as an initial datapoint to reflect the absence of parasite genes in the host prior to infection.

Results

Identification of candidate venom-like compounds

The overlap of all venom screening methods is displayed in Fig. 2. From the 114-peptide nematocyst proteome, 110 had structurally similar sequences in PDB or SCOP database, 48 of which had annotations containing target keywords (protease, inhibitor, toxin, venom). From this reduced set, 28 sequences contained complete, high-confidence (>95% Phyre2 confidence) structural domains and were retained for structural phylogenetics (Supplemental File 1). From the whole proteome, 26 sequences had BLAST homologs in the manually constructed venom database, 4 sequences had hits to HMMs, and 8 sequences were positive hits by ToxClassifier (Supplemental File 2). The de novo assembled infection transcriptome consisted of 42,543 transcripts with an N50 of 1,120 bp. The transcriptomic dataset we searched was 1/2–1/5 the size of those used by prior myxozoan venom studies (Foox et al., 2015; Hartigan et al., 2021) despite comparable sequencing depths. We used 100 bp single-end reads (whereas prior studies used paired-end), and we applied an additional level of contig filtering by annotation. A total of 884 sequences had BLAST homologs in our venom database, 71 sequences had venom-like domains predicted by HMMs, and 468 sequences were deemed positive hits by ToxClassifier (Supplemental File 2). HMMs with homologous sequences from each dataset are displayed in Fig. 3.

Figure 2 Venn diagrams showing candidate venom-like proteins identified by our screening methods.

(A) Proteins from the Ceratonova shasta nematocyst. (B) Transcripts from C. shasta infection transcriptome. Prepared with InteractiVenn (Heberle et al., 2015).

Figure 3 Results of HMM search of Ceratonova shasta nematocyst proteome and translated C. shasta transcriptome from infected fish, against 22 cnidarian venom HMMs from Klompen et al. (2020).

The HMMs in the outer box were not identified in C. shasta.

We identified two unique Kunitz-type protease inhibitor-like proteins based on similarity to sequences in Genbank. We refer to these as Kunitz-type protease inhibitor-like transcript 1, and Kunitz-type protease inhibitor-like transcript 2. In two cases, the C-type lectin-like protein and the peptidase inhibitor 16-like protein, identical, shorter sequences were recovered from the infection transcriptome. We selected the longer sequences from the nematocyst proteome for downstream analysis.

Altogether, 8 candidate VLCs passed our screening methods (Table 1). Those from the nematocyst proteome that were positive by at least three methods included a lactadherin-like protein, a peptidase inhibitor-16-like protein, and a C-type lectin-like protein. The infection transcriptome contained two metallopeptidase-like transcripts (astacin and reprolysin-like), two Kunitz-type protease inhibitor-like transcripts, a hyaluronidase-like transcript, and the same peptidase-inhibitor 16-like transcript and C-type lectin-like transcript found in the nematocyst proteome.

Table 1 Putative venom-like compounds identified in Ceratonova shasta datasets.

VLC name	Dataset	Length (aa)	Closest NCBI homolog	% Sequence identity	E-value	
C. shasta lactadherin-like protein	P	671	Cnidaria: Myxozoa
Myxobolus squamalis
KAF1744687	45.9	2e−75	
C. shasta peptidase inhibitor 16-like protein	P,T	825	Cnidaria: Myxozoa
Thelohanellus kitauei
KII69682	45.8	4e−56	
C. shasta C-type lectin-like protein	P,T	445	Cnidaria: Hydrozoa
Hydra vulgaris
XM_012708536	29.5	1e−41	
C. shasta hyaluronidase-like transcript	T	351	Chordata: Aves
Rhinopomastus cyanomelas
NXO00254	28.4	7e−43	
C. shasta Kunitz-type protease inhibitor-like transcript 1	T	119	Cnidaria: Anthozoa
Actinia tenebrosa
XP_031550075	47.8	6e−31	
C. shasta Kunitz-type protease inhibitor-like transcript 2	T	270	None			
C. shasta reprolysin-like transcript	T	193	Chordata: Actinopterygii
Silurus meridionalis
KAF7705091	28.9	7e−23	
C. shasta astacin-like transcript	T	198	Cnidaria: Myxozoa
Thelohanellus kitauei
KII65609	37.6	4e−33	
Note:

In the “Dataset” column, “P” indicates the compound was identified in the C. shasta nematocyst proteome; “T” indicates the transcriptome. Full amino acid sequences of these compounds are provided in Supplemental File 4.

Phylogenetic analysis of VLCs

We found homologs to the candidate C. shasta VLCs in all major classes of Cnidaria, including Polypodiozoa, Myxosporea, Malacosporea, Anthozoa and Medusozoa. We identified P. hydriforme homologs to 7 of our 8 C. shasta VLCs. For two, the reprolysin-like and peptidase inhibitor 16-like protein, we identified similar sequences in the malacosporean Tetracapsuloides bryosalmonae in transcriptomes from its invertebrate host but not fish host. For the astacin-like and Kunitz-type protease inhibitor-like transcript 2 transcripts, we found similar sequences in BLAST searches from venomous organisms outside Cnidaria (predominantly from class Arachnida). We included these sequences in the alignments and trees. We also included homologous sequence from the sponge Amphimedon queenslandica for all VLCs except the hyaluronidase-like transcript, for which a sequence was not available.

Maximum likelihood trees and corresponding alignments are displayed in Fig. 4 and Supplemental File 3. Several VLCs appear to be incomplete or are missing functional domains contained in related sequences. It is unknown whether this is an artifact of de novo transcriptome assembly or is biological reality. For three VLCs—the lactadherin-like protein, the C-type lectin-like protein, and hyaluronidase-like transcript—the topology of phylogenetic trees agree with taxonomy established via phylogenomic trees (Kayal et al., 2018; Chang et al., 2015). Five VLCs did not have topologies in agreement with taxonomy. These included the Kunitz-type protease inhibitor-like transcripts, the peptidase inhibitor 16-like protein, and both metallopeptidase-like transcripts.

Figure 4 Maximum likelihood phylogenetic trees and protein alignments from two VLCs with domains shown as colored blocks.

(A) Tree of C-type lectin-like protein has topology in agreement with established cnidarian taxonomy. (B) Tree of astacin-like transcript has topology incongruent with established cnidarian taxonomy and includes sequences from venomous animals outside of Cnidaria. See Additional File 3 for trees of all candidate VLCs.

Time-series expression of VLCs

More gene transcripts in the infection transcriptome were expressed later in infection (42% at 1 dpe, 46% at 7 dpe, 85% at 14 dpe, and >99% at 21 dpe). The largest single gene cluster identified by STEM, peaked at 1 dpe in the gills and decreased at later timepoints in the intestine, and included 30% of all genes. The nematocyst proteome homologs were expressed similarly (56% at 1 dpe, 60% at 7 dpe, 99% at 14 dpe, and 100% at 21 dpe). The largest STEM cluster included 27% of genes with expression peaking at 1dpe.

The three VLCs identified in the C. shasta nematocyst proteome matched highly similar sequences (tblastn e-value = 0) in the infection transcriptome. Of the eight candidates, three were expressed at 1 dpe (in gills) (Fig. 5). These include the C-type lectin-like protein, the hyaluronidase-like transcript, and the Kunitz-type protease inhibitor-like transcript 1. The five others had expression beginning at 14 or 21 dpe and were assigned to two other STEM clusters. These include the peptidase inhibitor 16-like protein, the lactadherin-like protein, the Kunitz-type protease inhibitor-like transcript 2 and the reprolysin and astacin metallopeptidase-like transcripts.

Figure 5 Heatmap of time-series expression of genes coding for the 8 VLCs, at 1, 7, 14 and 21-days post exposure (dpe).

There are three replicate fish per timepoint, each represented by a square that is shaded by gene read count, normalized for parasite replication. STEM Profiles are assigned to sequences by expression. Profiles with the same color belong to the same expression cluster.

Discussion

Our multifaceted approach yielded 8 VLCs in C. shasta, seven of which have been identified in other myxozoan species (Foox et al., 2015; Hartigan et al., 2021). These include C-type lectin-like proteins, metallopeptidases-like transcripts, peptidase-inhibitor-like proteins and Kunitz-type protease inhibitor-like transcripts. We found hyaluronidase-like transcripts for the first time in a myxozoan. Below, we discuss the individual C. shasta VLCs organized by postulated evolutionary origin; low bootstrap values on many of our VLC phylogenies limit the strength of support for these hypotheses.

“Inherited” VLCs

Phylogenetic trees of three C. shasta VLCs have topologies that parallel taxonomy, and thus we consider this as support for the hypothesis that some VLCs are inherited from a common cnidarian ancestor.

Lactadherin-like protein

Lactadherins are anticoagulants that compete with host blood-clotting factors (Shi & Gilbert, 2003). They are present in venom from snakes (Ching et al., 2012) and free-living cnidarians (Voolstra et al., 2017). The lactadherin-like sequence we identified had expression coinciding with the appearance of mature spores in histology at 21 dpe (Barrett & Bartholomew, 2021). The FA58C domains contained within this sequence is related to blood coagulation and has been noted in snake datasets (Ching et al., 2012; Rokyta, Wray & Margres, 2013; Junqueira-de-Azevedo et al., 2016). Potentially, lactadherin is translated late in infection to be packaged into mature nematocysts. While C. shasta tubules appear sealed, these venom peptides may be delivered in a mucus coating (Ramírez-Carreto et al., 2019) on the tubules, to suppress host blood clotting during initial attachment of the parasites.

C-type lectin-like protein

C-type lectins disrupt hemostasis and inflammation and are widely distributed in Cnidaria and other venomous animals (Morita, 2005). The C-type-lectin we identified is homologous to those recently identified in sea anemone transcriptomes (Klompen et al., 2020). In our infection transcriptome, C-type lectin expression peaked at 1 dpe in the gills. At 1 dpe, C. shasta is in an “amoeboid migration” phase, traveling via blood vessels from the site of entry to the intestine (Alama-Bermejo, Holzer & Bartholomew, 2019; Bjork & Bartholomew, 2010). At later timepoints, coinciding with C. shasta embedding in the intestine, this transcript had lower expression. Thus, we hypothesize that C-type lectin may help C. shasta evade the host innate immune response during initial migration (Loukas et al., 1999), and in the nematocysts, may serve a secondary, more typical venom function by disrupting host coagulation during attachment.

Hyaluronidase-like transcript

Hyaluronidases are regarded as “spreading factors” as they degrade hyaluronan to disrupt extracellular membranes, allowing the spread of toxins to local tissues (Tu & Hendon, 1983). They are present in the venom of many cnidarian taxa, including jellyfish (Lee et al., 2011), and sea anemones (Domínguez-Pérez et al., 2018). In C. shasta, we observed the hyaluronidase-like transcript was highly expressed at 1 dpe in the gills, and we propose that it may disrupt gill tissue to enable sporoplasm invasion. At 7 dpe, though expression was lower, hyaluronidase may facilitate the amoeboid “blebbing” motility described by Alama-Bermejo, Holzer & Bartholomew (2019), which could facilitate parasite penetration through gaps in host cellular matrices.

“Recruited” VLCs

Phylogenetic trees of the other 5 VLCs do not parallel established cnidarian topologies, despite all having sequence motifs characteristic of known venom components. We hypothesize that these compounds took on venom-like qualities via independent origins and convergent evolution, similar to what has been proposed with serine proteases inhibitors (Eszterbauer et al., 2020) and venom components (Hartigan et al., 2021) from other myxozoan taxa. We refer to these candidates as “recruited”. Recruitment of proteins for venom is common for venomous taxa including cnidarians (Jaimes-Becerra et al., 2017), platypus, and snakes (Whittington et al., 2008; Fry et al., 2009).

Peptidase Inhibitor16-like sequences

Peptidase inhibitor 16 (PI16) belongs to the CAP superfamily of proteins, and has been identified in venom proteomes (e.g. cone snails, Leonardi et al., 2012) and transcriptomes (e.g. scorpions, Cid-Uribe et al., 2018). Whereas its role in venoms is still unknown, in humans PI16 expression has been connected to conditions including heart disease (Frost & Engelhardt, 2007), cancer (Wang et al., 2020), and inflammation (Hazell et al., 2016). PI16 prevents the proteolytic activation of chemerin, a protein involved in recruiting macrophages to sites of inflammation (Regn et al., 2016). A PI16-like protein was discovered in the original description of the C. shasta nematocyst proteome (Piriatinskiy et al., 2017), and we recovered it again using our methods. The transcript encoding this protein was expressed at 14 and 21 dpe, while the parasite is proliferating. While the role of these proteins in free-living Cnidaria remains unknown, we hypothesize that in C. shasta infections, peptidase inhibitors are involved in host immune silencing to enable parasite replication, a phenomenon discussed in Barrett & Bartholomew (2021).

Metallopeptidase-like sequences

Metallopeptidases are proteolytic enzymes that utilize metals to cleave polypeptide bonds. They have various functions in organism development and are found in venom from snakes (Bjarnason & Fox, 1995), scorpions (Cid-Uribe et al., 2018) and cnidarians (Moran et al., 2013). In C. shasta, proteases are an essential aspect of virulence and vary greatly between genotypes (Alama-Bermejo et al., 2020). The two metallopeptidase VLCs we identified have different annotations and expression profiles and should be considered independently. Reprolysins are a subfamily of metalloproteinases known from snake venoms (Bjarnason & Fox, 1995). The reprolysin-like transcript was expressed at all timepoints and in both gill and gut tissue, where it may cleave host tissue for movement and proliferation. Astacins are membrane-secreted metalloendopeptidases in sea anemone venom. The astacin-like transcript was not expressed until 21 dpe. Like the lactadherin-like VLC, this late-expressed gene may be translated and packaged in nematocysts of mature myxospores and have a role in infecting the invertebrate host.

Kunitz-type protease inhibitor-like transcripts

Kunitz-type protease inhibitors are a family of protease inhibitors widespread in venomous taxa (Yang et al., 2014a), and identified previously in a myxozoan genomic dataset (Yang et al., 2014b). In sea anemones, Kunitz-type protease inhibitor proteins in nematocysts and mucus defend against proteolysis by predators and prey (Sintsova et al., 2018). Some parasites use Kunitz-type protease inhibitors to interfere with host immune response (Ranasinghe et al., 2015; Smith et al., 2020). We identified two Kunitz-type inhibitor proteins that have similar annotations but distinct expression profiles. Transcript 1 was heavily expressed at 1 dpe in the gills, and less at 7, 14 and 21 dpe in the intestines, whereas Transcript 2 was not expressed until 14 dpe. Asynchronous expression of protease inhibitors was likewise observed in M. cerebralis infections in Rainbow trout (Eszterbauer et al., 2021). Like these proteins in M. cerebralis, Kunitz-type protease inhibitors in C. shasta may have several roles in evading host immunity.

Sequences containing the ShK domain

Stichodacyla helianthus K -like (ShK) domain are small proteins that interfere with potassium channels. They are ubiquitous in sea anemone venom (Sachkova et al., 2020) and have structural orthologs in venomous organisms outside Cnidaria (Gerdol et al., 2019). Both of our datasets contained sequences with ShK-like domains detected by HMMER searches of the pfam database and venom HMMs from Klompen et al. (2020). While we are not prepared to ascribe these sequences as VLCs, they are notable targets for future investigation, as ShK domains also function in cnidarian nervous systems (Sachkova et al., 2020), which have not yet been observed in myxozoans.

Conclusions

We used a multi-faceted approach to identify putative VLCs in C. shasta proteomic and transcriptomic datasets. Relative to other cnidarians, VLCs in C. shasta are remarkably scarce: Similar approaches in free-living cnidarians have yielded ~100–600 candidate genes per species (Brinkman et al., 2015; Macrander, Brugler & Daly, 2015; Macrander, Broe & Daly, 2016; Lewis Ames et al., 2016; Klompen et al., 2020). However, we identified only 8, which are a subset of the 49 “toxins” identified in Myxobolus pendula (Foox et al., 2015), 124 in B. pumatellae and 96 from mixed myxosporean datasets (Hartigan et al., 2021). The 88–99% reduction in VLC diversity in C. shasta relative to venoms in free-living cnidaria parallels the 12–99% reduction in genome size in Myxozoa relative to free-living cnidarians (Adachi et al., 2017; Kim et al., 2019; Chang et al., 2015). Reduced VLC diversity in C. shasta may be an artefact of our different sequencing and assembly parameters and more-extensive manual screening.

Our novel time-series data suggest venom-like compounds function during different phases of the C. shasta infection of the fish host, corresponding with invasion, migration and replication of the parasite. Thus these VLCs do not fit the ‘predation and defense’ narrative of venom function in free-living cnidarians, instead, show that “venom” may have a diverse utility in parasitic organisms. The multi-faceted approach we used in this study should provide a means of identifying and describing venom components and other highly divergent proteins in Myxozoa. In addition, future studies could fluorescently label putative VLCs to determine their localization in myxosporean nematocysts, developmental stages, and mature spores to characterize atypical venom functions. Searches for VLCs in ‘omics datasets from myxozoan infections in corresponding invertebrate hosts may reveal multiple functions of the same compounds in alternate hosts of a complex life cycle. Given that we found a greater number of sequences similar to our VLCs in invertebrate infections of T. bryosalmonae, it may be that this venom reduction is less pronounced in the vertebrate-infective stages of myxozoans. Data that improve our understanding of the role of these compounds at the host-parasite interface during invasion and proliferation will simultaneously enrich the concept of what functions a “venom” can have and guide potential treatments for myxozoan infections.

Supplemental Information

Supplemental Information 1 Search results for venom-like compounds in Ceratonova shasta.

Nematocyst proteome and infection transcriptome sequences with results from venom search methods and time-series expression data in parasite-normalized transcripts per million counts.

Click here for additional data file.

Supplemental Information 2 Structural phylogenetics for 28/114 proteins in the C. shasta nematocyst proteome with structurally similar sequences whose functions contained the keywords: protease, inhibitor, toxin and venom.

A: Domain alignment visualizations with representative sequences from major tree branches. B: Phylogeny made with DALI for structurally similar sequences, whose functions contained the keywords. Yellow = sequence from database; Green = same clade; Purple = earlier common ancestor; Red = non-valid entry; Blue = not in tree.

Click here for additional data file.

Supplemental Information 3 Annotated phylogenies for eight venom-like compounds (VLCs).

Phylogenies and protein domain alignments for nine venom-like compounds and most closely related sequences. Protein domains were annotated using the NCBI conserved domains search tool.

Click here for additional data file.

Supplemental Information 4 FASTA file containing amino acid sequences for the eight venom-like compounds.

Click here for additional data file.

Bioinformatic analyses were performed on the computational infrastructure of the Oregon State University Center for Quantitative Life Sciences.

Additional Information and Declarations

Competing Interests

Author Contributions

DNA Deposition

Data Availability

The authors declare that they have no competing interests.

Benjamin Americus conceived and designed the experiments, performed the experiments, analyzed the data, prepared figures and/or tables, authored or reviewed drafts of the paper, and approved the final draft.

Nicole Hams conceived and designed the experiments, performed the experiments, analyzed the data, prepared figures and/or tables, authored or reviewed drafts of the paper, and approved the final draft.

Anna M. L. Klompen conceived and designed the experiments, analyzed the data, authored or reviewed drafts of the paper, and approved the final draft.

Gema Alama-Bermejo conceived and designed the experiments, performed the experiments, analyzed the data, authored or reviewed drafts of the paper, and approved the final draft.

Tamar Lotan conceived and designed the experiments, analyzed the data, authored or reviewed drafts of the paper, and approved the final draft.

Jerri L. Bartholomew conceived and designed the experiments, authored or reviewed drafts of the paper, and approved the final draft.

Stephen D. Atkinson conceived and designed the experiments, analyzed the data, authored or reviewed drafts of the paper, and approved the final draft.

The following information was supplied regarding the deposition of DNA sequences:

The sequences of the 8 VLCs, all sequences used in our analyses, and the amino acid sequences from only the 8 venom-like compounds referred to in the text and in Table 1 are available as Supplemental Files.

We use an already assembled draft genome referenced in Alama-Bermejo et al. (2020) DOI 10.1093/gbe/evaa109.

NCBI Assembly GCF_002163495.1.

The following information was supplied regarding data availability:

Raw data are previously published (as referenced in manuscript). We are using an already assembled draft genome referenced in Alama-Bermejo et al. (2020) DOI 10.1093/gbe/evaa109.

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
