# Peer review of "The cnidarian parasite Ceratonova shasta utilizes inherited and recruited venom-like compounds during infection"

_PeerJ, doi:10.7717/peerj.12606_

## Round 0.1 · original submission · Minor Revisions

Please address concerns of both reviewers and amend your manuscript accordingly.

Reviewer 1 ·

Basic reporting

no comment

Experimental design

no comment

Validity of the findings

no comment

Additional comments

In this study, the authors enumerate in the present study, that Cnidarians are the most ancient venomous organisms. They store a cocktail of venom proteins inside unique stinging organelles called nematocysts, and are present in all Cnidaria, including the morphologically simple Myxozoa, which are a speciose group of microscopic, spore-forming, obligate parasites of fish and invertebrates. The authors identified 9 VLCs, all of which have closely-related sequences in other myxozoan datasets, suggesting a conserved venom profile across Myxozoa, and an overall reduction in venom diversity relative to freeliving cnidarians.

The overall composition of the manuscript is good, is written in a clear and understanding manner with a very nice introduction to the topic. The quality of the figures and tables is also good. All used procedures are correct. The paper is scientifically and methodologically accurate, and the conclusions drawn are convincing. The linguistic style is sufficient for publication and the reference section is adequate. This manuscript will find the interest of many readers. The presented study can be proposed for publication when the authors check the following minor concerns:
Minor comments:
I suggest that the figure 1 should be described in the main text. The figure is not described in the main text.

Supplemental files need more descriptive metadata identifiers to future readers, and the figure should be improved using panels A, B, C. Besides, the authors should be providing figure legends for each figure described in the supplementary files.

Reviewer 2 ·

Basic reporting

The authors studied the venom-like proteins and transcripts of the myxozoan parasite, Ceratonova shasta. Using the already published C. shasta nematocyst (=polar capsule) proteome data, and the time-series transcriptome dataset obtained in the present study, the authors identified 9 venom-like compounds (VLCs) and detected their intrapiscine expression profiles from 1 to 21 days post exposure. Furthermore based on the outcome of phylogenetic analyses, the authors intended to predict if these VLCs are of cnidarian origin.

The MS is well written in general, although the wording is “lab-slang”-like sometimes (e.g. most heavily expressed). The figures are of adequate quality.

The main criticism is concerning the presentation and the discussion of findings. I suggest re-organizing completely the Results and Discussion parts. The Discussion contains results (and some sentences even refer to methods) which are not mentioned before, and the authors’ speculations regarding the putative functions of the detected VLCs are often not supported by the presented results. Therefore, the Results should be completed with all obtained data in detail, including the outcome of the structural modeling (mentioned in M&M L117-118). Please clarify if you obtained the full-length transcripts/proteins for predicting the structure. On the domain detection graphs (Figure 5 and Suppl. 3), some of the VLCs look partial (lactahedrin-like, metallopeptidase-like transcripts, Kunitz-type protein like ones). Please explain their status.

In M&M, the authors state that the outgroup of phylogenetic analyses is the poriferan Amphimedon queenslandica, but it is not the case for the hyaluronidase-like transcript. Please explain. In several cases, the low bootstrap values do not support the position of C. shasta transcripts (e.g. Figure 5 A, Suppl. 3 C-type lectin-like, lactadhedrin-like, Kunitz-type protein-like transcripts). I suggest discussing the issue, and please „soften” your conclusions regarding the inheritance of the examined proteins.

In the text, I would suggest discussing the “inherited” and “recruited” VLCs separately. Thus L248-253 would come after L300.

Some in-text citations are missing from the reference list (e.g. Hartigan et al. 2001, Eszterbauer et al. 2001). Please check thoroughly all references (both in text and in ref. list).

Experimental design

see above

Validity of the findings

see above

Additional comments

L96 Please check the sentence. Did you mean “…mapped to the C. shasta TRANSCRIPTOME…”?

Table 1 Please re-organize the table. In the first column, I would suggest adding an individual abbreviation for the examined VLCs that reflects both to the name of the organism (C. shasta) and the putative function of the analyzed transcript/protein (and then refer to this name throughout the MS). As the functions are predicted using in-silico, sequence-based analyses only, and the detected aa (??) sequence identities are rather low in most cases, I suggest mentioning in the annotation column that the identification is more a likeliness than a confirmed fact (i.e. C. shasta lactadherin-LIKE protein). Are the contigs obtained from Trinity RNA-seq assembler are available individually in Genbank? If not, please submit, and include here their Genbank Acc. No. Please add also the length of the contigs in Table 1.

Figure 4 What do the three squares per timepoint mean? Three parallels? Please explain in figure legend.

L219-221 Please be more precise with wording, name the referred proteins to help prospective readers to follow your thoughts (certainly, the same applies to the other parts of the MS).

L237 myxosporean

L246-248 multiple-times repetitions, please avoid

L251 serine protease INHIBITORs (serpins)

L252-253 Please add example (with reference) why you came to this conclusion.

L264-265 “We selected…” The sentence belongs to M&M. Use specific identifier, not a Trinity assembler code (the same applies to all transcripts, see the comment for Table 1).

L279 Do you mean over-expressed?

L295 at later timepoints (please add when exactly)

L326-327 It is result. Remove or re-word.

L341-341 “…not similar in sequence identity….”?? Please re-word. The correct reference here is Eszterbauer et al 2001, btw. How do you mean that the outcome of the two studies are comparable, if Kunitz-type proteins were not studied by Eszterbauer et al. 2021?

L347-348 Please explain why you think that Kunitz-type serpins are involved in evading host immunity. Could you add reference that support this theory?

L363-364 This is a crucial statement of the MS, therefore please explain in the previous parts how this 90% reduction was calculated. Is it possible that transcriptome sequencing was not complete, or during the filtering of raw reads, some potential VLCs were removed? Or your 4-step selection procedure was maybe too strict? Please discuss all these possibilities.

L368-374 repetition!

---

## Round 0.2 · accepted · Accept

All concerns were adequately addressed and the amended manuscript is acceptable now.

Reviewer 1 ·

Basic reporting

Professional article structure, figures, tables.

Experimental design

Rigorous investigation performed to a high technical & ethical standard.

Validity of the findings

Conclusions are well stated, linked to original research question.

Additional comments

The authors modified the observations adequately and the manuscript is substantially improved and can be accepted.